# Activation of Cryptochrome 4 from Atlantic Herring

**DOI:** 10.3390/biology13040262

**Published:** 2024-04-15

**Authors:** Anders Frederiksen, Mandus Aldag, Ilia A. Solov’yov, Luca Gerhards

**Affiliations:** 1Institute of Physics, Carl von Ossietzky University of Oldenburg, Carl-von-Ossietzky Straße 9-11, 26129 Oldenburg, Germany; anders.frederiksen@uni-oldenburg.de (A.F.); maldag@smail.uni-koeln.de (M.A.); ilia.solovyov@uni-oldenburg.de (I.A.S.); 2Research Centre for Neurosensory Sciences, Carl von Ossietzky University of Oldenburg, Carl-von-Ossietzky Straße 9-11, 26129 Oldenburg, Germany; 3Center for Nanoscale Dynamics (CENAD), Carl von Ossietzky University of Oldenburg, Ammerländer Heerstr. 114-118, 26129 Oldenburg, Germany

**Keywords:** electron transfer, radical pair mechanism, protein, magnetoreception, QM/MM, DFTB, multiscale model, cryptochrome

## Abstract

**Simple Summary:**

The Atlantic herring is one of many migratory fish that may use the geomagnetic field to navigate on its annual migration. The exact mechanism used for detecting the geomagnetic field in fish is still an open discussion, and the two main theories on magnetic sensing in animals are in the main focus: magnetite-based or radical pair-based. Here, we explore whether the cryptochrome 4 protein of fish would be able to carry out the necessary electron transfer activation to create a radical pair to be used for magnetic sensing.

**Abstract:**

Marine fish migrate long distances up to hundreds or even thousands of kilometers for various reasons that include seasonal dependencies, feeding, or reproduction. The ability to perceive the geomagnetic field, called magnetoreception, is one of the many mechanisms allowing some fish to navigate reliably in the aquatic realm. While it is believed that the photoreceptor protein cryptochrome 4 (Cry4) is the key component for the radical pair-based magnetoreception mechanism in night migratory songbirds, the Cry4 mechanism in fish is still largely unexplored. The present study aims to investigate properties of the fish Cry4 protein in order to understand the potential involvement in a radical pair-based magnetoreception. Specifically, a computationally reconstructed atomistic model of Cry4 from the Atlantic herring (*Clupea harengus*) was studied employing classical molecular dynamics (MD) and quantum mechanics/molecular mechanics (QM/MM) methods to investigate internal electron transfers and the radical pair formation. The QM/MM simulations reveal that electron transfers occur similarly to those found experimentally and computationally in Cry4 from European robin (*Erithacus rubecula*). It is therefore plausible that the investigated Atlantic herring Cry4 has the physical and chemical properties to form radical pairs that in turn could provide fish with a radical pair-based magnetic field compass sensor.

## 1. Introduction

The ability of migratory species to navigate safely on their long journeys has fascinated scientists for a long time [1,2,3,4,5,6,7,8]. Depending on the habitat, a wide array of sensory abilities is necessary to ensure reliable navigation. Among these abilities, many animals, like birds, fish, and insects use a magnetic compass for orientation during their long and often season-dependent journeys [8,9,10,11]. Despite extensive research, the precise mechanisms of how animals sense the geomagnetic field are still unknown while different theories have been proposed [3,12,13,14,15,16,17,18,19]. A promising hypothesis regarding how animals can navigate using the geomagnetic field was proposed by Schulten et al. in 1978, who suggested that the magnetic compass could rely on quantum mechanically entangled electrons [12]. The so-called radical pair mechanism (RPM) describes a non-equilibrium process where two molecules with an odd number of electrons form a correlated radical pair (RP). The possible spin states of the two unpaired electrons in an RP include the singlet and the triplet states. An external magnetic field (e.g., the geomagnetic field) may influence the interconversion between these states of the RP and consequently modulate the yield of a potential spin-dependent RP reaction product once it reaches thermal equilibrium. Changing the direction of the magnetic field affects the RP dynamics, thereby leading to different reaction product yields that may be neuronally processed [13,14,18,20,21].

Earlier studies have demonstrated that magnetoreception of migratory birds is blue light dependent [22,23] and the magnetoreception mechanisms are expected to rely on photosensitive molecules. In this regard, the photoreceptor protein cryptochrome 4 (Cry4), which non-covalently binds the chromophore flavin adenine dinucleotide (FAD), was shown to be a suitable candidate for avian magnetoreception found in the double cone cells in the bird’s retina [24,25,26]. Upon blue light excitation, the FAD has been shown to receive an electron from a nearby tryptophan (Trp_A_), creating the [${\mathrm{FAD}}^{.-}$Trp_A_^·+^] RP [24,27]. This fast electron transfer (ET) is followed by a cascade of intra-protein ETs between three further tryptophan residues terminating at a surface tryptophan (Trp_D_) and resulting in the [${\mathrm{FAD}}^{.-}$Trp_D_^·+^] RP (see Figure 1). It is believed that the [${\mathrm{FAD}}^{.-}$Trp_D_^·+^] RP is the most stable and has the potential to interact with the geomagnetic field [24,26,28].

The magnification of Figure 1 illustrates a zoom into a protein model of the Atlantic herring (*Clupea harengus*) Cry4 (ChCry4) and shows the FAD and the Trp_A–D_ tetrad with the described ETs. Important to note is that not only are the four principal tryptophans preserved between ChCry4 and avian Cry4 but the FAD binding site is also preserved. The preserved sites further indicate that on a sequence level the fish Cry4 has a high resemblance with the avian Cry4.

The goal of this study is to investigate the possibility of the existence of a RPM [13,24] in fish by investigating the electron transfer of the FAD-tryptophan chain of ChCry4. Since there are both studies suggesting magnetite particles [29] and the RP-based mechanism [30] as the primary magnetoreceptor in fish, the present study serves to explore if ChCry4 supports the underlying mechanism required for a RP-based mechanism to exist in Atlantic herring and, by extension, in fish. The existence of a RP-based mechanism in fish arose from a recent study which revealed that the Atlantic herring express Cry4 seasonally like avian species and the Cry4 from herring has a high structural and dynamical similarity when compared to the European robin Cry4a (ErCry4a) [30].

The present study relies on a structure of ChCry4 from an earlier investigation where it was reconstructed using homology modeling with pigeon Cry4 as the molecular template [30]. Employing the ChCry4 model, molecular dynamics (MD) simulations were performed to sample different starting structures for subsequent ET simulations. Finally, to investigate the conditions for successful electron propagations in ChCry4, hybrid quantum mechanics/molecular mechanics (QM/MM) simulations were performed. The results indicate that the studied model of ChCry4 supports the possibility of the protein to exhibit along ETs along the FAD-Trp chain.

## 2. Materials and Methods

MD simulations of the ChCry4 model were carried out using GROMACS [31] to create different starting structures for subsequent QM/MM simulations. The last frame of a previous NAMD [30,32] simulation was used as the starting frame in the new simulation. The GROMACS simulation was carried out using version 2022.4 and the Amber99SB-ILDN force field for proteins [33,34] with earlier parameterizations of the FAD cofactor [35,36]. The simulation box had a size of 103 Å × 104 Å × 113 Å. Firstly, the system was minimized using a steepest descent algorithm [37] where the minimization was stopped when the maximum force in the system was smaller than 1000 kJ/(mol·nm). After the minimization, the system was equilibrated in 2 steps by (i) a 2 ns NVT simulation in which Maxwell-distributed velocities were randomly assigned to the atoms corresponding to a temperature of 10 K and (ii) followed by a 1 ns NPT simulation where an isotropic Parrinello-Rahman algorithm was used for pressure control [38]. Finally, a 200 ns production simulation was carried out.

All simulation steps used an electrostatic and van der Waals-cutoff distance of 10 Å. The Particle Mesh Ewald (PME) method was used to treat long-range electrostatics and electrostatics across the periodic boundary conditions. Furthermore, the V-rescale method [39] was used as a thermostat with a reference temperature of 300 K while all covalent bonds were constrained. The equilibration and production steps employed a 2 fs integration step with a leapfrog integrator. A total of 51 equally distributed frames were sampled from the GROMACS MD simulation, each used as a starting structure for QM/MM simulations to describe electron hole transfers in the activated state of ChCry4.

To investigate the activated state of the ChCry4 model, the time-dependent RP populations in the system were modeled through real-time non-adiabatic QM/MM simulations. The Amber99SB-ILDN force-field was employed for the protein in the MM part of the system [33,34]. A density functional-based tight-binding (DFTB) method based on fragment molecular orbital implementation was used for the QM part. The QM part included the side chains of the four tryptophan amino acid residues within the FAD-Trp chain (Figure 1). The investigated tryptophan residues in the FAD-Trp chain included the amino acid residues 396 (Trp_A_), 373 (Trp_B_), 319 (Trp_C_), and 370 (Trp_D_). To model the activated state of the ChCry4, the FAD topology in the protein was changed to an anion-FAD variant carrying a negative charge which represents the state after an electron has been transferred from Trp_A_ to FAD. The simulations were performed in an in-house version of GROMACS [31] at a temperature of 300 K within an NPT ensemble employing the V-rescale method for temperature control and a Berendsen barostat at 1 bar for pressure control [39] with all bonds constrained. A time step of 1 fs as well as periodic boundary conditions and the PME method for long-range interactions were used for all 51 simulations spanning a 1 ns duration.

The QM/MM wave function of the transferring electron hole was recalculated at every time step and considered the protein and solvent environment such that a change in the environment due to the MD simulation was accounted for instantaneously. In reverse, the partial atomic charges of the tryptophan residues in the quantum region were updated accounting for the transferring electron hole. This approach permitted to account for the response of the environment to changes in the quantum region. The employed DTFB QM/MM method was used in earlier investigations with plant, amphibian, and avian cryptochrome [27,35,36].

## 3. Results and Discussion

The initial conditions of the QM/MM simulations assumed the first ET between the FAD and Trp_A_ to be completed, resulting in the initial RP state [${\mathrm{FAD}}^{.-}$Trp_A_^·+^]. This assumption was based on multiple earlier experimental studies that have shown a reliable transfer between
Trp_A_ and FAD in cryptochrome [27,40,41]. With time, the electron hole tunnels through the chain of tryptophan residues in ChCry4 (see Figure 1). An occupation of the electron hole at a certain tryptophan site Trp_X_ corresponds to a [${\mathrm{FAD}}^{.-}$Trp_X_^·+^] RP state where X = A, B, C, D. The temporal RP dynamics in ChCry4 can therefore easily be traced by calculating the population of the electron hole on the different tryptophan sites.

As follows from Figure 2A, the Trp_A_ occupation quickly decays from unity to population values below 0.1 within a 100 ps time interval, indicating that the initial RP state [${\mathrm{FAD}}^{.-}$Trp_A_^·+^] is rather unfavorable. Instead, the occupation of the RP state [${\mathrm{FAD}}^{.-}$TrpB^·+^] increases, making it the most favorable RP configuration as the corresponding occupation rapidly rises to 0.8 after 100 ps and continues to decay slightly to a value of about 0.7. Figure 2A shows that Trp_C_ has a low occupation over the entire time interval, indicating that this site rather acts as a bridge for the electron hole, connecting Trp_B_ and Trp_D_. Another argument for the role of Trp_C_ as a bridging site is given by the slow but steady rising occupation on Trp_D_, reaching a value of around 0.3 while the occupation on Trp_B_ decays. The simulations indicate that the [${\mathrm{FAD}}^{.-}$Trp_D_^·+^] RP is consequently the second most favorable state in the ChCry4 system.

The analysis showed that in 23 simulations an occupation of at least 0.3 is observed on Trp_C_ or Trp_D_. The ETs, which end with the electron on Trp_D_, are consistent with the earlier findings of the ETs in the European robin Cry4a according to Xu et al. [24], where the Moser–Dutton ruler [42,43] was used to show that the ETs are likely to complete within 1 ns. Furthermore, Xu et al. showed that the forward ET rate constants were orders of magnitudes greater than the backward rate constants in European robin Cry4a, which is in agreement with the presented results demonstrating the formation of the [${\mathrm{FAD}}^{.-}$Trp_D_^·+^] RP state.

The occupation data of the four RP states in ChCry4 permit estimating the characteristic ET rate constants. For this purpose, a numerical fit was applied to the average occupation of the RP state shown in Figure 2B, assuming a kinetic model that allows both forward and backward ET kinetics [35]:(1)$${\mathrm{Trp}}_{A}\underset{k_{f}^{AB}}{\overset{k_{b}^{AB}}{⇆}}{\mathrm{Trp}}_{B}\underset{k_{f}^{BC}}{\overset{k_{b}^{BC}}{⇆}}{\mathrm{Trp}}_{C}\underset{k_{f}^{CD}}{\overset{k_{b}^{CD}}{⇆}}{\mathrm{Trp}}_{D}\cdot$$

Here, $k^{AB}$, $k^{BC}$ and $k^{CD}$ denote the ET rate constants between two neighboring tryptophan sites while the indices *f* and *b* represent forward and backward transfer reactions, respectively. Note that the direction of the arrows corresponds to the simulated electron hole transfers; the corresponding ETs, illustrated in Figure 1, would have arrows in the reversed direction. The following equations define the kinetic model where A(t), B(t), C(t), and D(t), denote the time-dependent occupation of the RPA, RPB, RPC, and RPD states, respectively
(2)$$\begin{array}{c} \frac{dA(t)}{dt}=-k_{f}^{AB}A\left(t\right)+k_{b}^{AB}B\left(t\right) \end{array}$$
(3)$$\begin{array}{c} \frac{dB(t)}{dt}=k_{f}^{AB}A\left(t\right)-k_{b}^{AB}B\left(t\right)-k_{f}^{BC}B\left(t\right)+k_{b}^{BC}C\left(t\right) \end{array}$$
(4)$$\begin{array}{c} \frac{dC(t)}{dt}=k_{f}^{BC}B\left(t\right)-k_{b}^{BC}C\left(t\right)-k_{f}^{CD}C\left(t\right)+k_{b}^{CD}D\left(t\right) \end{array}$$
(5)$$\begin{array}{c} \frac{dD(t)}{dt}=k_{f}^{CD}C\left(t\right)-k_{b}^{CD}D\left(t\right) \end{array}$$

Figure 2B shows that the rate equations, Equations (Equation 2)–(5), could be used to interpret simulation results. It should be kept in mind that only a fraction of all simulations were used to estimate the transfer rate constants based on a similar approach utilized earlier by Timmer et al. to qualitatively obtain the completed ET rate constants [27].

The ET rate constants that follow from fitting the kinetic model, Equations (Equation 2)–(5), to the simulation data are given in Table 1 and provide the characteristic times required for an electron transferring from one site to another. Table 1 shows that the electron hole rapidly transfers from Trp_A_ to Trp_B_ while the forward transfer between Trp_B_ and Trp_C_ is an order of magnitude slower. However, if the electron hole reaches Trp_C_ it quickly jumps to Trp_D_ where the corresponding back transfer is less frequent with an average transfer time of 240 ps.

No experimentally obtained ET rate constants exist for ChCry4, but due to its similar structural and dynamical properties with ErCry4a [30] a comparison to experimental ErCry4a transfer rate constants is reasonable. Timmer et al. obtained ET rate constants for the two forward ETs with values of 30 ps for $1/k_{f}^{AB}$ and 141 ps for $1/k_{f}^{BC}$, both being in excellent agreement with the result shown in Table 1 [27]. It should be noted that the experimental rate constants were recorded at a temperature of 274 K compared to the 300 K employed for the present simulations. Unfortunately, no experimental rate constant exists for the last ET step from Trp_D_ to Trp_C_ or for the corresponding backward rate constants. The obtained ET rate constants show that the time for completing an ET is much shorter than the length of the simulations. The rate constants suggest that the cases where the radical pair remains on Trp_B_ do not come from a lack of simulation time but rather are due to other effects such as vibronic effects, which would require further studies to be elaborated.

A notable fraction of the QM/MM simulations revealed that the [${\mathrm{FAD}}^{.-}$Trp_B_^·+^] RP is very stable when compared to the same QM/MM approach used in earlier ET dynamics in *Arabidopsis thaliana* Cry1 (AtCry1) [35]. In this regard, the site energies $\epsilon_{i}$ of the Trp residues are important characteristics that influence the ET kinetics. Figure 3A–C illustrates that the site energies of certain Trp sites are the lowest for the simulations that have a high hole occupation at this site; the radical will rather populate a site with the lowest energy as it tends to be in the energetically most favorable state.

Another factor that influences RP evolution in protein is the electronic couplings $T_{ij}$, which are defined as the off-diagonal elements of the ET Hamiltonian. The electronic coupling is considered between any two tryptophan sites of the tetrad, though is only observed to be non-zero when considering two neighboring sites and is given in Figure 3D–F. Figure 3D shows that the electronic coupling is larger between Trp_A_ and Trp_B_ than between Trp_B_ and Trp_C_ (Figure 3E) or Trp_C_ and Trp_D_ (Figure 3F).

The higher electronic couplings correlate with the rapid decay of Trp_A_ occupation and the population of Trp_B_. The results in Figure 3D–F correspond to the electronic coupling values determined from all 51 performed simulations while Figure 3G–I shows the results for the 23 simulations where the transfer to Trp_C_ or Trp_D_ was completed. In the case when the electronic coupling should be responsible for ET completion, one would expect the $T_{ij}$ between Trp_B_ and Trp_C_ and between Trp_C_ and Trp_D_ to be larger as compared to the values in cases when the ET was stuck at Trp_B_. The results in Figure 3H and Figure 3I show that there is no notable change when compared to Figure 3E and Figure 3F, respectively. This result possibly suggests a less important role of the electronics couplings $T_{ij}$ for the ET rate constants and emphasizes the influence of the site energies as clear correlations between site energies and electron hole occupations are observed. The difference in internal energetics of ChCry4 could be attributed to the influence of the solvent on the radicals since it was previously shown in QM/MM simulations of AtCry1 that the solvent drastically influences the formation of RPs in plant cryptochrome. Lüdemann et al. [35] observed that if the solvent was not taken into account, the [${\mathrm{FAD}}^{.-}$Trp_A_^·+^] RP would be strongly stabilized, suggesting that the other ET steps would be hindered energetically. Lüdemann et al. concluded that the solvent acts as a driving force for the ETs in the case of AtCry1. The number of water molecules within 5 Å of each of the 4 tryptophan residues in ChCry4 was calculated for (i) the simulations where the ETs stop at Trp_B_ (stuck) and (ii) for the cases where the radical propagates to Trp_C_ and Trp_D_ (completed).

Figure 4 shows that the number of water molecules around the Trp residues do not significantly differ for the stuck and completed cases (see Figure 4A,C,D), except for the case around Trp_B_ for which the number of water molecules is greater with the radical being stuck on Trp_B_ (see Figure 4B). The difference in the number of water molecules around Trp_B_ grows with the simulation time ending with a difference of at least one extra molecule in the case when the electron hole is stuck on Trp_B_. Although the difference in the number of water molecules is clear, there is significantly less water around Trp_B_ compared to the two near-surface residues Trp_C_ and Trp_D_. Nevertheless, it is arguable that a high number of water molecules around Trp_B_ stabilizes the RP [${\mathrm{FAD}}^{.-}$Trp_B_^·+^] by screening the Coulomb interaction to the other tryptophan sites, which then might even lead to more water molecules around Trp_B_. This conclusion of a stabilization of the RP state is due to the higher number of water molecules, combined with the result from Lüdemann et al. [35]. This suggests that the solvent is important for ETs and that the completed ETs happen with a certain number of water molecules around the tryptophan sites. A too-high or too-low number of water molecules could, therefore, lead to energetically unfavorable configurations of the protein and ETs become less likely.

The important role of water involvement in ETs might be connected to the opening of the phosphate-binding loop in the activated state of the protein. Earlier studies performed on pigeon Cry4 demonstrated that a residue number between 220 and 240 may significantly rearrange upon protein activation leading to a greater solvent-accessible surface area around the FAD [44,45]. The opening of the phosphate binding loop in the activated state could work as a solvent control mechanism influencing the ET processes.

Another characteristic that could affect the ET is the edge-to-edge distance between Trp_B_ and Trp_C_. One would expect that in the completed simulations the distance between Trp_B_ and Trp_C_ is smaller as the electron could then be transferred more rapidly. Figure 5 shows the calculated distribution of the edge-to-edge distances with a Gaussian fit for the simulations where the electron hole was stuck on Trp_B_ (black) and for the simulations that completed the electron hole transfer along the FAD-Trp chain (red). The results turn out to be the opposite of expectations as the inter-residue distances are slightly greater for the completed ETs; the maxima of the Gaussian fitting curves appear at 4.2 Å for the completed and 4.0 Å for the stuck ETs. The distance is usually a determining factor for ETs the transfer rate constants depend exponentially on the donor acceptor distance in the widely applied Marcus theory [42,46,47,48]. The larger distances in the completed ET cases suggest that the reason for the preferred state in ChCry4 does not exclusively rely on the edge-to-edge distances. Instead, when considering the similarity in the electronic couplings $T_{ij}$ seen in Figure 3D–I and the increased distance for completed ETs, the observed blocking of ET is suggested to be a consequence of increased solvent exposure. Besides that, only 51 simulations were performed due to the utilized resource demanding approach, leaving the possibility for statistical artefacts. The attachment of the electron hole to the Trp_B_ residue along with its unstable behavior at the Trp_C_ residue may both diminish if a greater set of calculations is considered. However, the present investigations only focused on the possibility of an ET, which was clearly demonstrated in Figure 2.

## 4. Conclusions

A computational approach was presented to investigate the activation of Atlantic herring cryptochrome 4 and the formation of RPs. The simulations reveal different behaviors of the electron hole dynamics. In all 1 ns-long simulations, the electron hole transferred from the starting site on Trp_A_ to Trp_B_; 28 simulations showed the electron hole stuck on Trp_B_, while 23 simulations had an occupation of at least 0.3 on either Trp_C_ or Trp_D_, corresponding, respectively, to the formation of the [${\mathrm{FAD}}^{.-}$Trp_B_^·+^], [${\mathrm{FAD}}^{.-}$Trp_C_^·+^] and [${\mathrm{FAD}}^{.-}$Trp_D_^·+^] RPs. The ETs from Trp_B_ to Trp_A_ were robustly observed after less than 100 ps.

In the observed ChCry4 system, the [${\mathrm{FAD}}^{.-}$Trp_B_^·+^] RP was the formed routinely, as shown by Timmer et al. in ErCry4a, employing the same methodology. A possible physical reason for the [${\mathrm{FAD}}^{.-}$Trp_B_^·+^] RP formation could be attributed to the lowest energy of Trp_B_ compared to the other sites, as predicted within the approximation utilized in this study. On the other hand, the analysis suggests that a higher number of water molecules around Trp_B_ could lead to energetically favorable protein configurations, making the ETs highly dependent on the solvent. A continued in-depth study concentrating on the environmental and vibronic effects on the charge transfer dynamics might reveal more insights. It is noteworthy that the theoretical ChCry4 ET rates are similar to the experimental ErCry4a rates despite different temperature regimes.

With the [${\mathrm{FAD}}^{.-}$Trp_D_^·+^] RP formation, we argue that ChCry4 has the appropriate properties required for the RPM to exist in Atlantic herring and correspondingly play a role in magnetoreception. It should be kept in mind that with the available information this conclusion remains somewhat speculative and depends on additional verification of the magnetoreception mechanism in fish.

## Figures and Tables

**Figure 1 biology-13-00262-f001:**
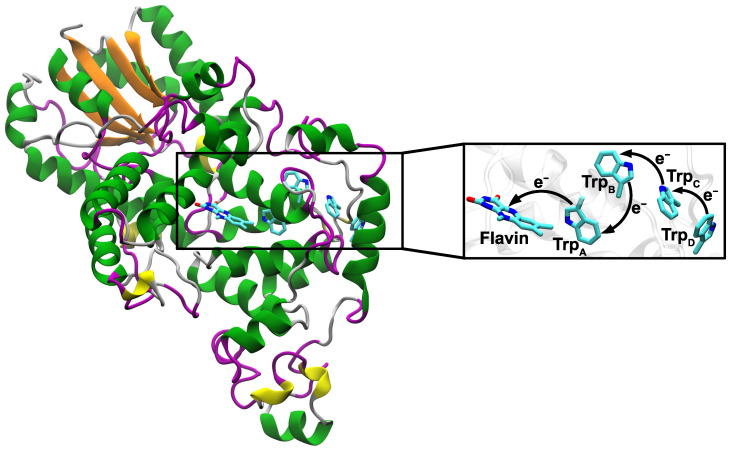
Illustration of the studied Atlantic herring Cry4 model with a blow-up of the region around the FAD cofactor showing the electron transfer cascade in the FAD-tryptophan chain that is crucial for the RPM. Only the heavy atoms (no hydrogen) of the flavin in the FAD and the heavy atoms of the sidechains of the tryptophan residues are shown. Here, Trp_A–D_ denote the conserved tryptophan residues 396, 373, 319, and 370 in the protein.

**Figure 2 biology-13-00262-f002:**
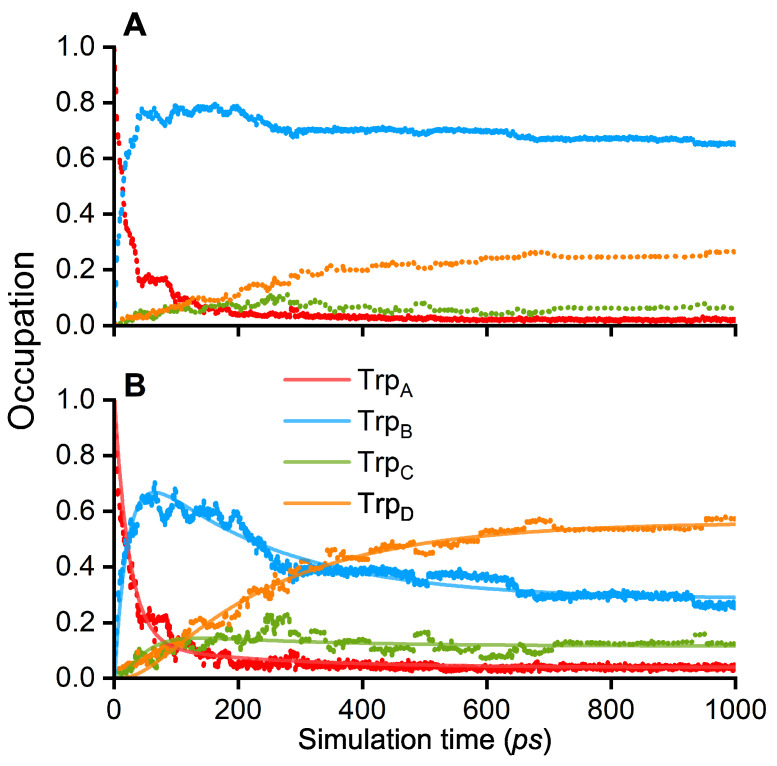
Occupations of the tryptophan sites Trp_A_, Trp_B_, Trp_C_ and Trp_D_ with an electron hole plotted over an interval of 1000 ps (points). (**A**) averaged over 51 QM/MM simulations; (**B**) averaged over the 23 simulations where an occupation greater than 0.3 was observed at the Trp_C_ or Trp_D_ at some time instance. The solid lines represent numerical fits of the averaged data obtained by the kinetic model in Equations (2)–(5).

**Figure 3 biology-13-00262-f003:**
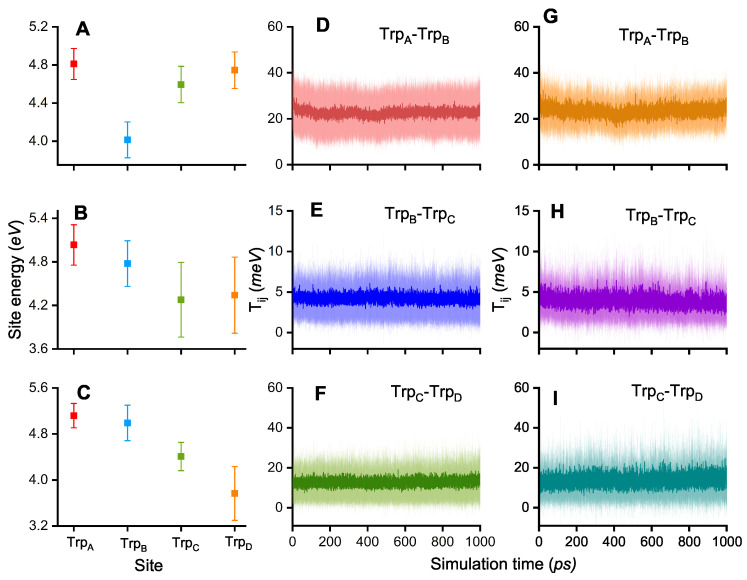
(**A**–**C**) Site energies $\epsilon_{i}$ of the four Trp residues in the FAD-Trp-chain, where *i* = A, B, C, D colored red, blue, green and orange respectively, for three simulations with different outcomes: Simulation outcome with high electron hole occupation on Trp_B_ (**A**), Trp_C_ (**B**), and Trp_D_ (**C**). Average electronic couplings $T_{ij}$ with *i* and *j* denoting two neighboring Trp sites A, B, C, or D, and the color corresponds to the acceptor site, calculated over (**D**–**F**) all 51 simulations and (**G**–**I**) over the 23 simulations with an occupation of Trp_C_ or Trp_D_ greater than 0.3. The average electronic couplings are plotted over the simulation time in dark color with the corresponding standard deviation in light color.

**Figure 4 biology-13-00262-f004:**
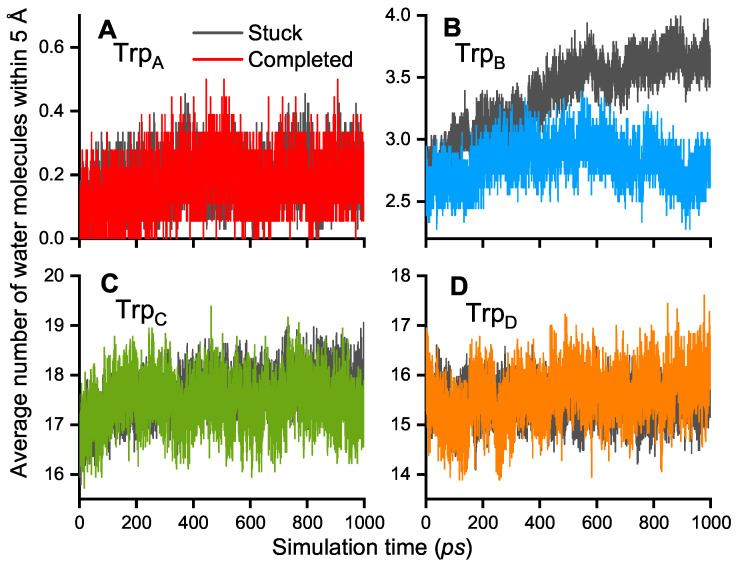
Average number of water molecules within 5 Å of each Trp residue within the FAD-Trp chain of ChCry4. (**A**–**D**) Black lines: average over all 28 simulations with the radical stuck on Trp_B_ for each site. Colored lines: average over the other 23 simulations where the Trp_C_ or Trp_D_ became radicalized, corresponding to the color-scheme used in Figure 2.

**Figure 5 biology-13-00262-f005:**
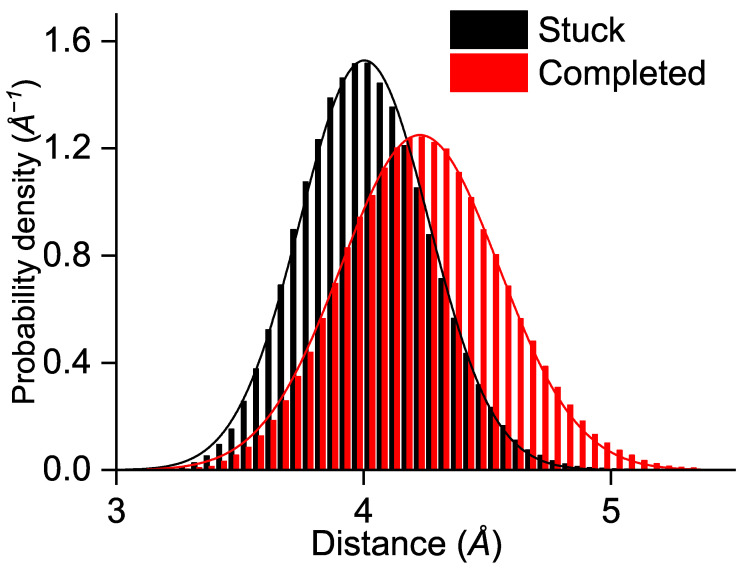
Edge-to-edge distance distributions between Trp_B_ and Trp_C_ in the QM/MM simulations with Gaussian fitting curves. Black: the results from the simulations with the electron hole stuck on Trp_B_. Red: the results from simulations where the electron hole propagates to Trp_C_ or Trp_D_.

**Table 1 biology-13-00262-t001:** Time constants (inverse of the corresponding transfer rate constants) computed for the ET chain in ChCry4.

1/*k* (ps)	Trp_A_–Trp_B_	Trp_B_–Trp_C_	Trp_C_–Trp_D_
$1/k_{f}$	23	149	57
$1/k_{b}$	141	82	240

## Data Availability

Data are available upon request by mailing the corresponding author.

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
