# Peer review of "Activation of Cryptochrome 4 from Atlantic Herring"

_biology, 2024, doi:10.3390/biology13040262_

Round 1

Reviewer 1 Report

Comments and Suggestions for Authors

The manuscript used a computationally model of Cry4 from the Atlantic herring (Clupea harengus) to study the internal electron transfers and the radical pair formation by employing classical molecular dynamics (MD) and quantum mechanics/molecular mechanics (QM/MM) methods. The simulations reveal that electron transfers occur similarly to those found experimentally and computationally in Cry4 from European robin (Erithacus rubecula). Although the manuscript has a good background of previous study, some points should be addressed before it can be recommended for publication.

1. There are a lot of punctuation or phrase errors throughout the text, which have been highlighted in the attached document and need to be corrected by the authors.

2. In Page 8, line 246-258, what does it mean with “The attachment of the electron hole on the TrpB along with the unstable behavior on TrpC could both diminish if a greater set of calculations is performed”?. Also, “only the possibility of an ET was focused on….”? Please make the description more clarified.

3. The conclusions are too long. Should be condensed or refined.

Comments on the Quality of English Language

There are a lot of punctuation or phrase errors throughout the text, which have been highlighted in the attached document and need to be corrected by the authors.

Author Response

We refer to the attached pdf file for detailed answers of the reviewers comments.

Reviewer 2 Report

Comments and Suggestions for Authors

This is an useful, professionally performed and generally well presented work. My only recommendation is to lead the reader more friendly towards the conclusion that "ChCry4 is expected to have appropriate properties to play a role in magnetoreception in fish". The possible role of ChCry4 in magnetoreception and navigation in fish is the main motivation for this work, as seen from abstract and introduction. However, it is not easy to jump to this conclusion from simulated kinetics of electron hole transfer in flavin and triptophane residues. Some more extended explanations, let them be qulitative, seem to be in place here.

Comments on the Quality of English Language

The English is good as I can judge, not being a native speaker. However, the authors should double-check for minor errors and misprints which I have noticed here and there throughout the text. Just one example: in lines 58-59 "the Atlantic herring expresses Cry4 seasonally like avian species and has a high structural and dynamical resemblance when compared to European robin Cry4a" - quite obviously, the herring bears no recemblance to Cry4a, something is missing here.

Author Response

(The authors gave the same response as above.)

Reviewer 3 Report

Comments and Suggestions for Authors

In the manuscript ‘Activation of Cryptochrome 4 from Atlantic Herring’ its authors, Anders Frederiksen, Mandus Aldag, Ilia A. Solov’yov and Luca Gerhards, report the results of their computational study of Cry4 from the Atlantic herring (Clupea harengus) employing classical molecular dynamics (MD) and quantum mechanics/molecular mechanics (QM/MM) methods to investigate internal electron transfers and the radical pair formation. The team from the University of Oldenburg has recently reported the presence of time-compensated sun compass in juvenile Atlantic herring [Ref. 4], and they have apparently found magnetic compass orientation in this species [Ref. 30]. It’s a pity that the latter reference is currently ‘submitted’ to another journal, i.e. is not available to the reviewer. However, with the current push for rapid publication it is understandable that the authors submitted this contribution before the paper on the magnetic compass was published.

This is a very sound study that shows that that ability of herring Cry4 to form radical pairs [FAD·– TrpD·+] in the simulations on a similar timescale as avian Cry4a, as well as the high structural and dynamic resemblance of herring Cry4 to avian Cry4a (which is shown in the manuscript that is not available to the reviewer), herring Cry4 is expected to have appropriate properties to play a role in magnetoreception in fish. Certainly the magnetoreception mechanism used by juvenile herring needs to be verified, but this contribution is an important first step.

Minor remarks:

Lines 51-52: ‘the aquatic Cry4 has a high resemblance with the avian Cry4’. It seems stylistically odd to compare ‘aquatic’ and ‘avian’ Cry4. I suggest comparing fish and avian cryptochromes.

Line 266: no need for a comma after ‘however’.

Author Response

(The authors gave the same response as above.)

Round 2

Reviewer 1 Report

Comments and Suggestions for Authors

The authors have considered and addressed all the comments raised in previous edition. I am glad to see that the revised manuscript is now suitable for publication in the present form in Biology.